# Rain and small earthquakes maintain a slow-moving landslide in a persistent critical state

Noélie Bontemps[1]*, Pascal Lacroix [1], Eric Larose [1], Jorge Jara[2] & Edu Taipe [3]

In tectonically active mountain belts, landslides contribute significantly to erosion. Statistical analysis of regional inventories of earthquake-triggered-landslides after large earthquakes (Mw > 5.5) reveal a complex interaction between seismic shaking, landslide material, and rainfall. However, the contributions of each component have never been quantified due to a lack of in-situ data for active landslides. We exploited a 3-year geodetic and seismic dataset for a slow-moving landslide in Peru affected by local earthquakes and seasonal rainfalls. Here we show that in combination, they cause greater landslide motion than either force alone. We also show the rigidity of the landslide's bulk clearly decreasing during $Ml \geq 5$ earthquakes. The recovery is affected by rainfall and small earthquakes ($Ml < 3.6$), which prevent the soil from healing, highlighting the importance of the timing between forcings. These new quantitative insights into the mechanics of landslides open new perspectives for the study of the mass balance of earthquakes.

[1] Univ. Grenoble Alpes, Univ. Savoie Mont Blanc, CNRS, IRD, IFSTTAR, ISTerre, 38000 Grenoble, France. [2] Laboratoire de Géologie, Département des Géosciences, École Normale Supérieure, CNRS, UMR 8538, PSL Research University, Paris, France. [3] OVI-INGEMMET, Arequipa, Peru. *email: noelie.bontemps@univ-grenoble-alpes.fr

L large and shallow earthquakes (Mw > 5.5) trigger widespread landsliding in mountainous areas[1–3], thus contributing to the mass balance of earthquakes. Most of the earthquake-triggered landslides are activated co-seismically by moderate to large earthquakes (magnitude > 4)[4]. Various mechanisms have been invoked to explain the co-seismic triggering of landslides: dynamic loading due to seismic shaking increases the shear stress applied to the sliding surface, and thus the shear forces can overcome the shear resistance[5–7]. The multiple ground motion cycles progressively weaken the soil by fracturing the rock mass[8]; eventually, if associated with high precipitation, small displacements along the sliding surface could also generate a rapid drop in shear resistance due to the crushing of the grains in the shearing plane. This reduction in volume can lead to a rapid increase in pore pressure if the sliding surface is in the water table, and hence potentially a co-seismic reactivation and/or a rapid triggering of the landslide[9,10]. This last mechanism known as undrained loading was observed in both fine grain rock[10–12] and soil slides[9,13], and can lead to the reactivation or the triggering of landslides.

In addition, some observations indicated different time scales in the delay between seismic shaking and landslide activation[1,14], often when the earthquake was combined with precipitations, the other main forcing of landslides. At the scale of a week, a change in groundwater conditions (pore pressure and/or permeability increase) can occur due to microfractures and cracks in the bedrock caused by the shaking[15]. The resulting changes in water content can take several days to reach the landslide's sliding surface, hence the landslide can occur several days later[14,16]. Increased rates of rapidly triggered landslides were also observed in various regions, months to years after major earthquakes (Mw > 6.6)[17]. Slow-moving landslides can also be affected over the same time scales, with increased velocities observed in the years following a Mw = 5.4 earthquake[18]. A possible mechanism for these changes, which is corroborated by field surface observations, involves the decrease of rock strength due to earthquake-generated micro-/macrofractures[17]. These fractures can then generate preferential paths for precipitation infiltration[17,19] and/or increase hillslope failure due to the decreased mechanical strength of the rock[19–21]. The different time scales raise questions as to the relationship between shaking, precipitation, material damage, and/or permeability. The time scales of the healing processes in highly weathered media, where rainfall is usually the main factor triggering landslides[22,23], are also currently unknown. Current hypotheses as to the physical processes behind how this earthquake-precipitation combination triggers landslides are based only on qualitative observations[1,14,17,19]. We still lack quantitative observations at the landslide-scale to confirm the mechanisms involved. In this study, we aimed to document those mechanisms using local information such as surface displacement and material rigidity, obtained from in situ measurements of a slow-moving landslide in an environment where earthquakes may be combined with seasonal differences in rainfall.

This study highlights the importance of the timing between earthquakes and precipitation, together with the role of small and medium magnitude earthquakes (Ml < 4.5) in landslide kinematics and the recovery process of the soil's rigidity. We point out the damage of the soil generated by strong seismic shaking and explains the long-term impact of medium to large earthquakes on landslide triggering. Our results also show how small-shaking events can affect the landslide rigidity when combined with precipitations. They will indeed prevent the recovery of the rigidity of the soil with time.

## Results

**Context and experimental setup.** The broad slow- to very slow-moving landslide of Maca (Fig. 1 and Supplementary Fig. 1) has been identified has a clay/silt compound slide with a rupture surface of uneven curvature[24] and thus, belongs to the soil slide category[25]. This 60 million m³ landslide impacts a village in a rural area in the Colca Valley, southern Peru, in a very seismically active zone[26,27]. This area also experiences seasonal precipitations falling entirely between December and May[24] (Fig. 2a). The first 8–12 m of the landslide are composed of a very permeable layer from a 10,000-year-old debris avalanche, whereas the underlying material corresponds to fine lacustrine deposits with a thickness >50 m. In the fastest part of the landslide, the sliding surface is located at least 40 m down from the surface, within the lacustrine deposits[24].

The landslide velocity estimated over the last 10 years is about 1.5 m.yr$^{-1}$ in its fastest zone[18], and may reach up to 7 m.yr$^{-1}$ during strong rainy seasons[18,24] (Fig. 1a). The kinematics of the landslide is mainly driven by precipitations and by the river eroding its toe, but there exists a rainfall threshold above which deformation is triggered[24]. Regional earthquakes (Mw ≤ 6) can also accelerate its motion in a co-seismic way, followed by a period of relaxation[5]. The landslide kinematics is therefore affected by both earthquakes and rainfall, but the mechanics of the material in the bulk of the landslide remains uncertain.

The persistence of the motion allows monitoring different physical parameters of the landslide's mechanics and their evolution with time. As a consequence, its unique situation in an area where both rainfalls and earthquakes combine, makes it of high interest to study the landslide's mechanics under seismic/rainfall forcings. The kinematics of the Maca landslide is representative of a broad range of landslides, showing typical seasonal behaviors[24,28–30], as well as the existence of a rainfall threshold to initiate the motion[30–33] and similar pattern of motions than other landslides of the valley during earthquakes[34]. Its characteristics are typical of landslides in lacustrine deposits, in terms of velocity, geology, thickness, and mechanisms[35–39]. For all these reasons, this landslide has been intensively monitored since 2011.

In December 2015, a hut hosting a GPS and a broadband seismometer was installed on the fastest part of the landslide (Fig. 1a, Supplementary Figs. 1 and 2). These two instruments continuously monitor two geophysical parameters: the surface displacement of the landslide and the relative seismic velocity changes in the landslide body (dv/v) using ambient seismic noise[40–42]. This latter parameter is related to variations in soil density and/or rigidity[41,43]. In addition, GPS campaigns have been performed at the hut's location every 3 months since 2013, and were used to measure the overall landslide displacement between 2013 and 2015 at this specific location[24].

**Combined effect of rainfall and earthquakes on landslide dynamics.** We first focused on the surface GPS displacement and the seismic activity in rainy and dry seasons.

Over the period of study, high seismic activity was recorded, with 165 Ml 3.1 to 5.5 earthquakes within a 50 km radius of the landslide. Two earthquakes significantly affected the landslide's kinematics over the period studied. The first one occurred on the 20 February 2016, during the rainy season, and the second on 15 August 2016 during the dry season. The magnitudes of these earthquakes were estimated by the Peruvian Geophysical Institute at 5.0 and 5.5, respectively, on the local magnitude scale (Ml), and their sources were located at 9 km and 11 km from the landslide, respectively (see Fig. 1b). The ground shaking caused by the August

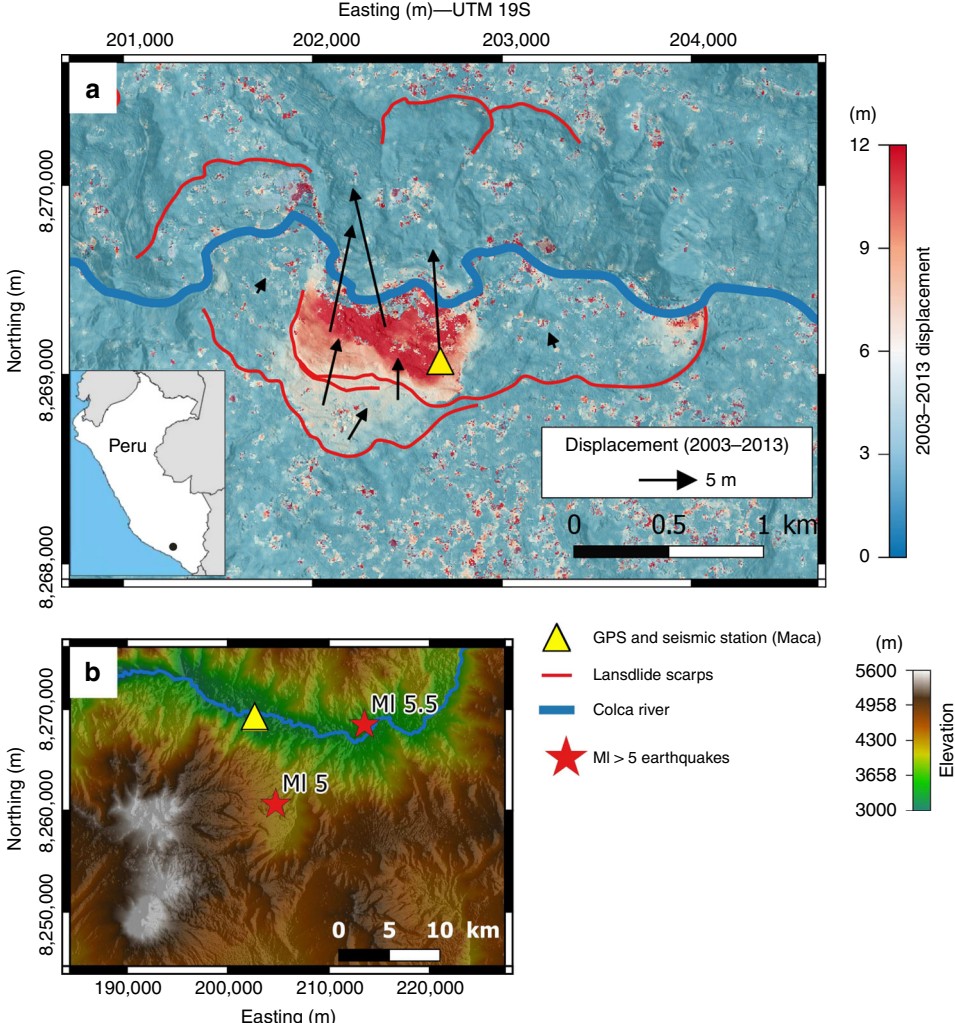

**Fig. 1 The Maca landslide. a** Cumulative displacement of the Maca landslide between 2003 and 2013 (from Bontemps et al.[18]). **b** Digital elevation model of a part of the Colca Valley. The red stars represent the epicenters of the Ml 5.0 and the Ml 5.5 earthquakes that occurred on 20 February and 15 August 2016, respectively. The yellow triangle indicates the location of the hut housing the GPS and seismic station on the Maca landslide.

2016 earthquake on the landslide was significantly more extensive than that caused by the February 2016 earthquake (approximated Peak Ground Velocity $PGV_{app}$ of $5.3 \, cm.s^{-1}$ compared with $3.5 \, cm.s^{-1}$ for the February earthquake—see Methods section).

As a result of the 15 August 2016 earthquake, the Maca landslide underwent a co-seismic slip of 1 cm followed by a relaxation period that lasted 30–40 days, during which the cumulated displacement was 11 cm (Fig. 2b, red curve). These co- and post-seismic effects corroborate previous observations of the same landslide following a Mw6.0 earthquake, which also occurred during a dry season, the origin of which was located 20 km from the landslide[5]. After the February earthquake, the Maca landslide underwent co- and post-seismic slippage of 80 cm over 5 months. This displacement is much greater than that recorded after the August earthquake even though the estimated shaking generated at the landslide's location by the February earthquake was weaker than in August. This observation suggests that rainfall plays a major role in how an earthquake triggers/ activates a landslide, and hence reveals a strong combination between the two different forcing events.

To validate this hypothesis, we must eliminate the possibility that the rain alone could have generated equivalent landslide displacements during the 2016 wet season. To do so, we focused

our attention on the 2014 and 2017 rainy seasons and the displacements observed over these periods.

The 2014 rainy season was comparable to the 2016 season in terms of precipitation recorded (Fig. 3). However, seismic activity was much lower in 2014, with only 44 earthquakes with a PGV > $0.01 \, cm.s^{-1}$, compared with 99 in 2016 ($0.01 \, cm.s^{-1}$ corresponds to the minimum PGV measured on the landslide among all the local earthquakes detected by the IGP network in the Colca Valley). Interestingly, the landslide displacements measured in 2014 were almost zero. Moreover, the rainfall recorded in 2017 (a year once again with much lower seismicity than 2016) was more than twice that recorded for 2016 (see Fig. 3). Nevertheless, the displacements in 2017 were only half those recorded during the 2016 rainy season. This difference in displacements between the different wet seasons can therefore be attributed to the greater seismic activity in 2016 than in 2017 or 2014 (Supplementary Fig. 10). Thus, our results represent the combined effect of earthquake shaking and precipitations on the acceleration of a slow-moving landslide, similar to what has been observed with rapid landslides triggered following large earthquakes[17]. These observations indicate that the combination of earthquakes and precipitations leads to greater landslide motion than either phenomenon alone. Hereafter, we take advantage of the

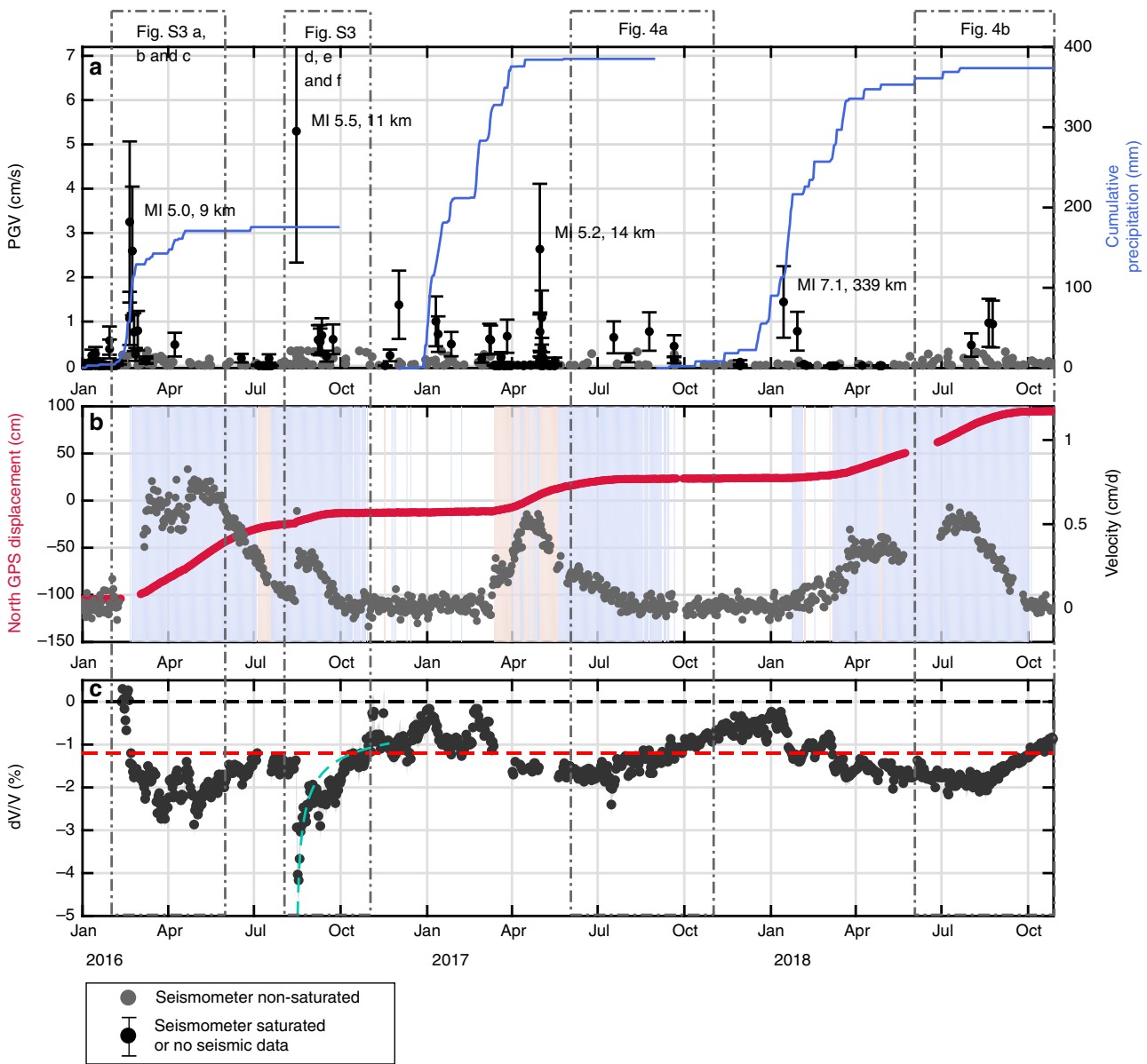

**Fig. 2 Comparison of landslide displacement and dv/v induced by earthquakes and precipitation. a** Seismicity reported by the Peruvian Geophysical Institute (IGP) for earthquakes with PGV above 0.01 cm.s$^{-1}$ (see SI) on the landslide (gray dots), and cumulative rainfall over the rainy season (solid blue line). Earthquakes strong enough to saturate the seismometer located on the landslide or earthquake where no records were available due to problems with the seismometer are indicated by black dots. For significant earthquakes, magnitudes and distances to the Maca landslide were indicated. **b** Time series of the Maca GPS North cumulative displacement (red) and velocity (gray) at a 1-day sampling rate. Blue zones represent periods were the dv/v in under −1.2%, and red zones represent periods were the dv/v is likely under −1.2% but gaps in the seismic data exist. **c** Changes to relative seismic velocity of the material determined by comparing daily seismic noise correlograms in the 3–8 Hz frequency range (black), which have a depth sensitivity of up to 40 m (see Supplementary Figs. 12 and 13). Gaps in the dv/v time series correspond to missing records. The horizontal black dashed line is the surface wave velocity on the first day of our study period. The red horizontal dashed line is an empirical threshold used in the text (−1.2%) under which the landslide is in a critical state. The logarithmic green curve represents the recovery function proposed by Richter et al.[48]. Boxes indicate part of the graphic that has been enlarged either in Supplementary Fig. 3 or in Fig. 4. Close-up of the years 2017 and 2018 are displayed, respectively, in the Supplementary Figs. 4 and 5.

persistence of the slow-moving landslide movement to study the mechanism producing this combined effect.

**An Ml=5 earthquake causes considerable damage to the unstable mass.** We investigated the existence and causes of any soil alterations, based on measurements of the soil's rigidity over time thanks to ambient seismic noise correlation methods[41,43] (see Methods section).

Following the Ml 5.5 August 2016 earthquake, the surface wave velocity in the 3–8 Hz frequency range dropped co-seismically by

more than 2% (Fig. 2c and Supplementary Fig. 3f). Investigation of the sensitivity of the surface waves at different frequencies and depths indicated that the 3–8 Hz range corresponds to a depth sensitivity of ~40 m (see Methods section and Supplementary Figs. 12 and 13). The variations observed are therefore located in the landslide's body.

We examined the influence of three different mechanisms, i.e., undrained loading[10], the variation of the water table[44], and the damage of the soil due to earthquake shaking[17], upon the dv/v in the 3–8 Hz frequency range, thanks to poroelastic models[45,46].

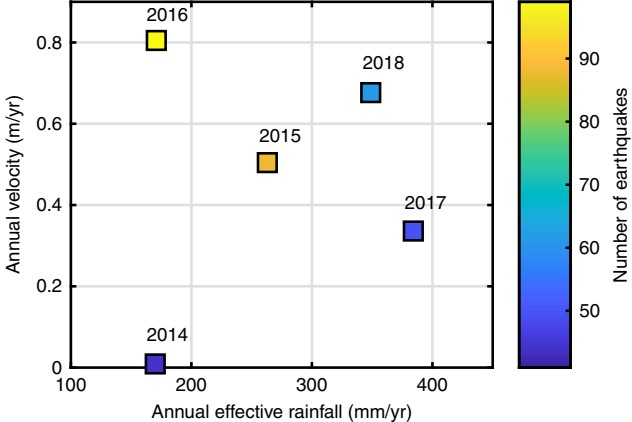

**Fig. 3 Annual landslide velocity as a function of effective annual rainfall.**
Colors indicate the number of earthquakes with PGV at the landslide
location above 0.01 cm.s$^{-1}$. See SI and Supplementary Fig. 6 for more
information. The effective annual rainfall corresponds to the cumulative
effective precipitation between December until mid-August of each year.
The cumulative displacement for each year corresponds to the total
displacement between the first and the last date of the rainy season were
the landslide's velocity is above a certain threshold (set at 0.039 m.yr$^{-1}$).
Earthquakes are counted only in between 2 weeks prior to the acceleration
of the landslide, up to when the motion ceases in each rainy season. The
displacement and seismicity following the August 2016 earthquake are not
taken into account for 2016. These periods are summarized in the
Supplementary Fig. 6 and Supplementary Note 1.

The effect of undrained cyclic loading[10] was estimated by
modeling the liquefaction of the sliding surface, i.e., a strong
reduction of the S-wave velocity near the sliding surface, between
38 and 40 m (see Supplementary Fig. 14a). Our model predicts
strong dv/v variations at low frequencies, with a maximum
variation of ~−3% around 3 Hz (Supplementary Fig. 14b). Our
observations of the dv/v variations at different frequencies the day
of the August 2016 earthquake do not correspond to the model
predictions (Supplementary Fig. 14b). Indeed the observed drop
is overly low around 3 Hz and too high above 5 Hz to be
explained by undrained loading (Fig. 2 and Supplementary
Figs. 14b, h).

At first sight, the possible effects of variations in water table
level on the observed dv/v could be neglected as the earthquake
occurred in the dry season, several months after the last
precipitations, implying a lower saturation of the soil. However,
to be certain, we modeled a 1-m elevation of the perched aquifer
at 10-m depth (see Methods section). This modeling could only
explain a dv/v drop of 0.04% i.e., 2 orders of magnitude below the
dv/v drop observed between 3 and 8 Hz (Fig. 2 and Supplemen-
tary Fig. 3).

The co-seismic drop in velocity can therefore be attributed
predominantly or even completely to damage to the soil[45,47–49],
due to the opening of new or pre-existing cracks[47,48,50–52].

Almost immediately after earthquake shaking, the dv/v was
seen to recover. This recovery was clearly visible, for instance,
with the August 2016 earthquake (Fig. 2c and Supplementary
Fig. 3f). This healing phase can be interpreted as the re-
compaction of the soil as fractures close, and the grains cement
together, reflecting a viscoelastic response of the soil[47,48,50–52].
This slow healing was also observed at laboratory-scale, where it
is referred to as slow dynamics[53]. Previous observations of the dv/
v recovery after an earthquake show that it often evolves
logarithmically over time[47,48,51]. A similar pattern was also
observed here, with recovery of the pre-earthquake soil state
around one and a half months after the Ml 5.5 earthquake in

August 2016 (Fig. 2c, light-green logarithmic curve and
Supplementary Fig. 3f). However, recovery is questionable during
the wet season, as we cannot separate it from the drainage
process. This becomes even more complicated when small-
earthquake activity is high, as we will see in the next sections.

**Recovery during wet seasons reveals the mechanism causing
the combined effect.** To better understand the combined role of
water and earthquakes, we focused on the Ml 5.0 February 2016
earthquake, which occurred during a rainy season (Fig. 1c and
Supplementary Fig. 3c).

The drop in dv/v for this earthquake was smaller than that
measured in August 2016, possibly due to the difference in
shaking intensity between the two events. However, in contrast to
the event of August, the dv/v continued to decrease for several
weeks after the Ml 5.0 event. As the surface wave velocity depends
on the rigidity of the soil, we interpreted the co-seismic drop
observed as soil damage due to the strong shaking, just like
during the August earthquake (see Supplementary Fig. 14b). The
difference in dv/v behaviors in the weeks following the earth-
quake can be explained by a transient increase in soil density and/
or decrease in rigidity due to an augmentation of the water
content in the lacustrine deposit. Indeed, the damage to the
different ground layers, generated by the strong shaking
associated with the February earthquake could have promoted
water infiltration at depth. These observations provide quantita-
tive measurements of the damage process which was qualitatively
reported in a previous study[17].

In addition, the movement of the landslide in itself favors the
formation of cracks and fissures in the soil, which produce or
amplify preferential water infiltration paths[54–57]. Following water
infiltration, the activity of the landslide can be self-sustained. As a
result, the combined effect of earthquakes and rainfall is amplified
on slow-moving landslides.

**Effect of small ground shaking on the landslide dynamics.**
Several questions can be raised related to the conditions (shaking
limits and rainfall amounts) activating this damage/recovery
process. To address these questions, we will now focus on periods
with small to moderate seismic shaking, i.e., from June to end-
October of 2017 and 2018 where 121 events were recorded, all
with PGV inferior to 1 cm.s$^{-1}$ (Fig. 4).

During the period of observations, we noted that almost every
time the dv/v dropped below a certain threshold (empirically set
at −1.2%), the landslide entered into motion (Fig. 2c, red
horizontal line), i.e., was in the "critical regime." Once the dv/v
recovered and exceeded this threshold once again, the landslide
slowed down and eventually stopped, entering the "stable
regime."

Interestingly, the recovery phase, defined by the time at the end
of the rainy season when the dv/v started to return to baseline,
occurred later in 2018 than in 2017 (Fig. 4a, b). More precisely, in
2017, the dv/v recovered from mid-July, several months after the
last precipitations, as the unstable mass dried out and its rigidity
increased (Fig. 4a). In contrast, recovery in 2018 only started at
the end of August, 1 month later than the previous year (Fig. 4b).
This difference in dv/v recovery could neither be attributed to a
difference in cumulative precipitation (Fig. 2a), nor a delay in the
rainy season (both rainy seasons were almost finished by the end
of April), nor due to large seismic events (the number of events
with large Peak Ground Velocity (PGV) was even higher in 2017).
The Ml 7.1 that struck the Peruvian coast in January 2018, does
not appear to have caused the difference either, as it occurred at
the beginning of the wet season and no co-seismic variations of
the surface wave velocities were observed at the seismic station.

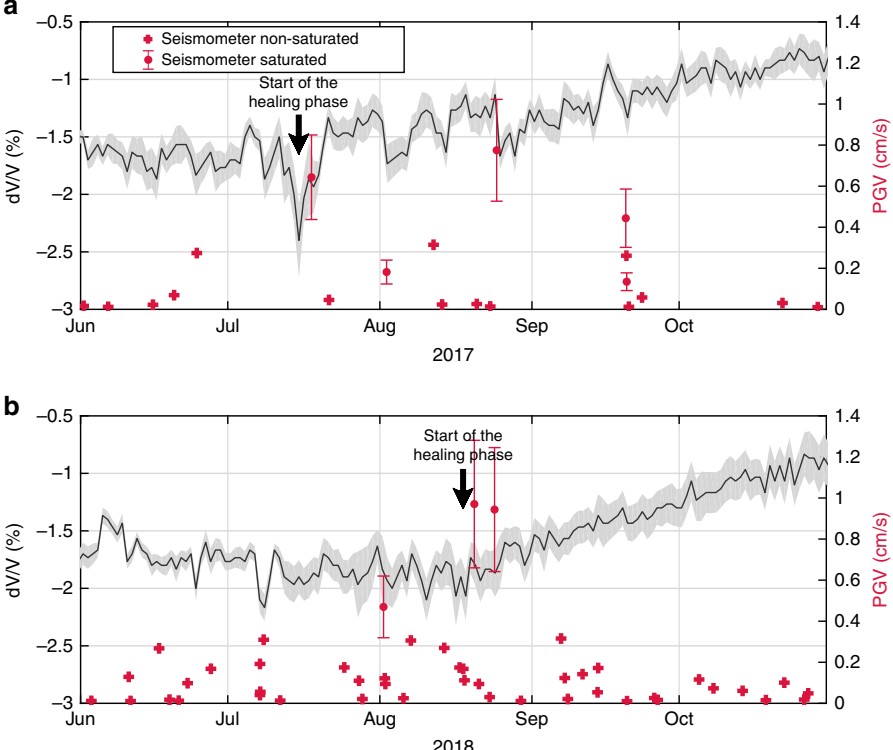

**Fig. 4 Comparison of the recovery of the dv/v at the end of the 2017 and 2018 wet seasons.** dv/v variations in the landslide material, obtained by comparing daily seismic noise correlograms in the 3–8 Hz frequency range in 2017 (**a**) and 2018 (**b**), together with all earthquakes reported by the IGP institute in Peru that have a PGV recorded at the landslide site above 0.01 cm.s$^{-1}$ (cross mark) in 2017 (**a**) and 2018 (**b**). In addition, the PGV of earthquakes strong enough to saturate the seismometer placed on the landslide were estimated (dots) (see Methods for details).

The only remaining difference between 2017 and 2018 is the rate of low-magnitude earthquakes, which was greater at the end of the rainy season in 2018 than in 2017. Indeed, 30 seismic events of low ground motion (PGV<1 cm.s$^{-1}$) were recorded between June and the end of August 2018, compared with only 13 earthquakes in 2017 during the same period (see Fig. 4). This difference suggests that the recovery phase in 2018 was delayed due to the intense seismic activity at the end of the rainy season. Hence, while water is stored in the landslide body, even small earthquakes can alter the landslide kinematics by limiting recovery processes, and thus keeping the landslide in the critical regime.

Note that at the end of August 2018, two large earthquakes with a PGV estimated close to 1 cm.s$^{-1}$ occurred. This PGV is higher than any previously recorded in or estimated for 2018. Nevertheless, these earthquakes had no influence on the dv/v and did not prevent the initiation of the recovery phase (Fig. 4b). At the end of August, the water content in the landslide was lower than between May and July due to sparse and low precipitation after the month of May 2018 (Fig. 2a). We therefore hypothesize that the impact of the low PGV on the landslide's rigidity decreases when water content in the soil also drops, eventually allowing the dv/v to increase and initiation of the healing phase. Three mechanisms can be invoked to explain this small-earthquake effect. The shaking can break recent/weak bonds between the elements of the highly unconsolidated granular material (i.e., clay, siltstone, shale), which include chemical and capillary bonds[58,59]. It can also generate fluid circulation and alter pore water pressure during the wave train, or slightly alter the arrangement of unconsolidated grains. These three effects, individually or combined, can maintain the landslide in its critical state.

## Discussion

Our results show that the combination of earthquakes and rainfall affects the rigidity of the Maca landslide. These observations allowed us to schematize the mechanics of a slow-moving land-slide exposed to rainfall and earthquakes (Fig. 5): once the rigidity of the landslide drops below a defined threshold, deformations and displacements are triggered and the landslide enters a critical regime. Precipitations maintain the rigidity of the landslide below the threshold. In addition, in the critical regime, landslide dis-placement may also induce additional damage and loss of rigidity, meaning that the landslide is also partially self-sustaining. Once the rain-related water infiltration stops, the healing phase can start and landslide displacement will come to an end until the next forcing capable of triggering displacements (see Fig. 5 for a summary).

Our results also show how small-shaking events (PGV < 1 cm.s$^{-1}$ corresponding mainly to Ml < 4.5 in this region) can alter landslide rigidity and hence, its kinematic behavior over time. Indeed, small events combined with precipitation can maintain the low rigidity, preventing the unstable mass from recovering. Possible causes of this effect include prevention of the creation and/or breaking of existing chemical and/or capillary bonds between the elements of the highly unconsolidated granular material[58,59], rearranging the grains, and/or increasing the pore pressure. To summarize, the landslide remains in a critical regime for longer than if small earthquakes had not occurred, as its rigidity stays below a certain critical threshold. In this context, the density of earthquake occur-rence is a key parameter determining whether the landslide remains in a critical state.

The result of this study not only confirm previous observations relating to the combined effect of rainfall and earthquakes on either slow-moving landslides[60], rapid landslides[17] or rockfall

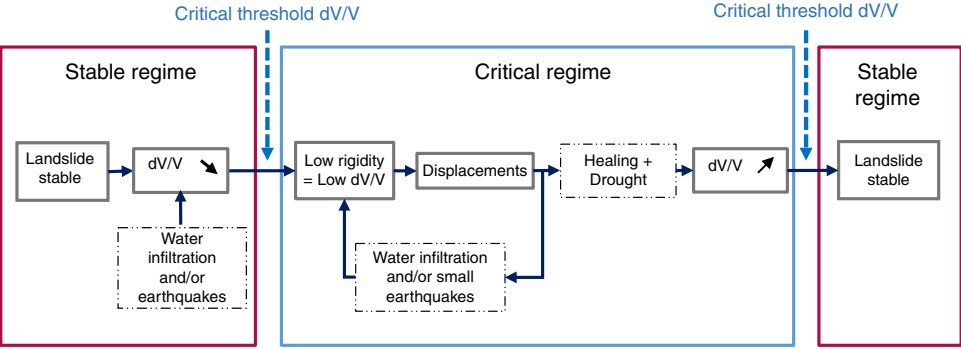

**Fig. 5** Schematic representation of the different forcings affecting the slow-moving Maca landslide.

activity[61], but also provided quantitative data based on the dv/v which support the previous hypothesis that soil damage causes this combined effect[17]. Soil damage, interpreted as the creation of micro-/macrofractures as a result of earthquake shaking, generates preferential paths for water infiltration and thus impacts landslide motion after an earthquake until the soil has completely healed. This mechanism would also explain the observations of rainfall threshold variation observed throughout the world after medium to large magnitude earthquakes[18,62,63].

Even though these observations were obtained on a single landslide owing to the uniqueness of the site in terms of monitoring techniques and to the difficulty to maintain such monitoring over the years, previous measurement at this site showed the representativeness of the Maca landslide in terms of kinematics during earthquakes[34] and seasonal rainfall forcings[24,28–30]. The Maca landslide is also similar to other landslides with regard to its geology, velocity, thickness, and mechanisms[35–39]. Furthermore, the processes highlighted here (damage and healing) are observed at large scale[47,48] and in all types of materials in laboratory experiments[53,64]. For these reasons, the mechanisms highlighted here are not specific to the Maca landslide but provide important information on the earthquake/rainfall combination on landslides in general. We should note, however, that quantitative differences should certainly exist between different landslides with different geology, thickness, and pore saturation.

These results help to better understand landslide forcing mechanisms and the strong combination between earthquakes and precipitations. This analysis has considerable implications both for landslide prediction and for the long-term impact of earthquakes on landscape evolution. The findings presented here support the idea that medium-intensity earthquakes (Ml < 5.5) can also accelerate landslides, and that their impact will be greater when precipitations are recorded before or just after the event. These observations also highlight the strong impact of cumulated small earthquakes (Ml 3.2–3.6) combined with a high water content that prevents the landslide from recovering its rigidity, maintaining the unstable mass in a critical regime and enhancing displacements. This observation underlines the importance of the temporality between precipitations and earthquakes on triggering landslides, and consequently on the mass balance of earthquakes. Finally, this study also highlighted the importance of considering smaller earthquakes and slow-moving landslides when determining the mass balance of mountain building.

## Methods

**GPS processing**. Fifty-five continuous GPS (cGPS) recordings were processed from various networks in the Andean region, South America and Nazca Plates (see Supplementary Fig. 7 in Supplementary Information). Thirty-three stations were installed and are maintained in Southern Peru and Northern Chile as part of the following projects: Integrated Plate Boundary (IPOC, www.ipoc-network.org), LIA-MB (www.lia-mb.net), CAnTO (www.tectonics.caltech.edu), CSN (www.csn.uchile.cl), and CAP

(www.unavco.org). The remaining stations are part of the International GNSS Service (IGS, www.igs.org) global network. Data were processed using the double-difference method in GAMIT 10.6[65], choosing the ionosphere-free combination and setting the ambiguities to integer values. Precise orbits from IGS, precise EOPs from the IERS bulletin B, IGS tables describing the phase center of the antennas, FES2004 ocean-tidal loading corrections, and atmospheric loading corrections (tidal and non-tidal) were used. One tropospheric Zenith delay parameter was estimated every 2 h, and one pair of horizontal tropospheric gradients were determined per 24-h session using Vienna Mapping Functions (VMF1)[66]. These estimations allow the tropospheric delay to be mapped in the perpendicular direction, with a priori ZHD derived from pressure and temperature values from VMF1 grids. The daily solutions and position time series were obtained using PYACS software[67], mapped first onto the ITRF 2008 reference frame[68] and then onto the South American plate, using a fixed pole[69,70].

**Landslide velocity computation**. To reduce noise when computing the velocity of the landslide at the location of the Maca GPS station, the displacement was averaged over a 5-day moving window. However, to reduce the smoothing effect and provide more realistic velocities after the August 2016 earthquake, the velocity was not smoothed for the 2 days following the earthquake.

**Seismic data and ambient seismic noise processing**. A Noemax © seismic station was used for this study. It has three components and a natural frequency of 4.5 Hz. This seismometer includes an electronically-compensated instrumental correction allowing a relatively flat response between 0.1 Hz and about 50 Hz.

The ambient noise in the frequency bands studied in Maca was mostly caused by vehicles on the main road located near the landslide, the river at the toe of the landslide and wind. These sources of noise can be considered stable in time only when the 24-h correlations are averaged, computed over each day as explained hereafter.

Single station cross-correlation was performed on the three possible pairs of components[51], i.e., North-East, vertical-East, and Vertical-North. To do so, hour-long records were spectrally whitened between 0.2 and 35 Hz to equalize the frequency content of the ambient noise[43]. The seismic signal was then normalized in amplitude using the clipping method such that the signal-to-noise ratio did not exceed three times its standard deviation. The main objective of this step was to attenuate the statistical weight of large events, such as earthquakes, compared with other events making up the ambient noise. The hour-long records were then correlated for the three possible pairs of components separately.

The next step was to filter the correlograms to enhance the signal-to-noise ratio by applying a Singular Value Decomposition-based Wiener filter[71]. The parameters used for this filter were $K = 7$, $L = 7$, the filter orders applied to the vertical and horizontal dimensions, respectively, and the number of first singular values was $N = 30$. Finally, correlations were averaged over each day in order to increase the signal-to-noise ratio and to eliminate dv/v sub-daily variations due to source variations during the day. However, even though the clipping reduces the impact of earthquakes on the ambient noise correlation, they will slightly modify hourly correlations and thus daily correlations by acting as noise sources that are not constant in time. The direct consequences will be a decorrelation of the daily correlations with regard to the reference during the stretching step and thus an increase of 30%, on average, of the dv/v uncertainties compared with days with less seismicity.

Daily correlations were then compared with each other to track relative velocity changes using the stretching method[72,73] applied to each component pair separately over the frequency band 3–8 Hz. The reference value for each stretch was obtained by averaging daily cross-correlation over the whole period of study. Stretched parts of the coda were selected between [−14Δt:2Δt] and [2Δt:14Δt] seconds, i.e., [−2.8: −0.4] and [0.4: 2.8] seconds in the 3–8 Hz frequency band, to remove the autocorrelation of the noise source which has a duration $\Delta t = 1/\Delta f$. Here, we removed $2 \times \Delta t$ to be certain that the source noise is not used in the stretching step (Supplementary Fig. 8). Consequently, analysis mainly focused on

scattering of surface waves (Rayleigh and possibly Love waves) in the strongly heterogeneous medium.

To decrease the signal-to-noise ratio, the dv/v was averaged over the three directional correlation pairs. The absolute error of dv/v was estimated from the correlation coefficient obtained by the stretching method[74].

**Surface wave depth sensitivity**. As surface waves are mainly composed of shear waves (S waves), active seismic acquisitions on the landslide were used to measure the velocity of the S waves. The surface wave inversion technique was applied in the Geopsy software (http://www.geopsy.org) with signals generated by dropping a mass of 80 kg from a height of 2 m using a tripod (see Supplementary Fig. 11). The model obtained by inversion gave a first layer of ~8–12 m with S waves traveling at around 110 m/s. Below, lies another layer composed of lacustrine deposits, where the S-wave velocity was close to 350 m/s (see supplementary Note 2 for further details).

From this model, it was possible to study the Rayleigh wave sensitivity as a function of depth and frequency using the gpdc package available in the Geopsy software. Perturbation of the Rayleigh wave velocity observed at the surface was examined by imposing a decrease of 14% of the shear wave velocity over a 50-cm layer at different depths to the reference model (Supplementary Fig. 13 and supplementary Note 3). The surface waves filtered between 3 and 8 Hz were estimated to be sensitive to depths down to 40 m, i.e., within the unstable mass of the soil.

**Velocity of P and S waves as a function of saturation**. In order to model the impact of the variation of the water table height upon the surface wave velocities, we used the Biot-Gassmann equations[45] as done in other studies[46,75,76]. These equations give the velocity of Vp and Vs as a function of the porosity and their fluid saturation. The velocity of the lacustrine deposits previously obtained thanks to the geophysical survey (see SI), i.e., $Vp = 1900$ m.s$^{-1}$ and $Vp = 350$ m.s$^{-1}$, were used as fitting values to set the equations parameters (the porosity and the consolidation coefficient) and calibrate them when the soil is completely saturated. As no values of the bulk and shear modulus were obtained on the landslide, we used as input the bulk and shear modulus of the feldspar plagioclase, one of the main minerals present in the landslide[77], i.e., $Ks = 75.6$ MPa and $Gs = 25.6$ GPa[78]. We fit the P- and S-wave velocities by choosing a porosity of the soil of 35% and a compaction coefficient of 120. Then the gpdc package available in the Geopsy software is again used to estimate the Rayleigh wave velocity variations as a function of the frequency for undrained loading (see Supplementary Fig. 14b and the supplementary Note 4) or when a water table at 10-m depth experiences a 1-m variation. This last model predicts a variation of around 0.04% between 3 and 8 Hz. The sign of the variations will depend on the direction of the water table variation.

**Earthquake ground motion approximation**. The seismic shaking intensity was estimated by computing the PGV, which is one indicator of seismic ground motion. The PGV can be computed directly from the signal measured by the seismometer installed on the landslide. However, a large number of earthquakes with magnitudes ranging between Ml3.5 and Ml7.1 saturated the signal registered by the seismometer. Consequently, the PGV was not always accessible, in particular for the largest and/or nearest earthquakes. For those events, the PGV was computed from Ground Motion Prediction Equations (GMPE). However, no GMPE was available for Peru alone[79] and most GMPEs are based on moment magnitude, whereas the magnitudes given by the Peruvian Geophysical Institute correspond to local magnitudes. Two classical GMPEs[80,81] were therefore compared and the fit between the estimated and the measured PGV was determined for unsaturated signals. The used GMPEs were developed for earthquakes with magnitudes ranging between Mw5 and Mw7.6 for the first model[81] and between Mw4 and Mw8 for the second one[80]. Their maximum distance validity are, respectively, 100 and 400 km. We chose to push the limit of the valid magnitude for both models by comparing measured and estimated PGV for earthquakes between 3.5 and 8, and to see their efficiency in our area when using local magnitudes instead of moment magnitudes. Concerning the maximum distance, we used all earthquakes that did not saturate the landslide within a radius of 400 km from the landslide, corresponding to the maximum distance of the second GMPE[80]. These limits allowed us to compare PGV values from a catalog of 348 events between February 2016 and November 2018.

Finally, the PGV of all these events was computed for the two horizontal components: North ($PGV_N$) and East ($PGV_E$). The final measured PGV is defined as follows: $PGV_{measured} = \sqrt{PGV_N * PGV_E}$.

The fit between measured an estimated PGV (see Supplementary Fig. 9), shows a better result using the GMPE from Akkar and Boomer[81] for our dataset. Therefore, the PGVs were taken directly from the seismic signal when the seismometer did not saturate, and from the Akkar and Boomer GMPE when the Maca station saturated.

The PGV was considered to have no uncertainties when determined from recorded seismic signals. However, the uncertainty on the PGV obtained from GMPEs could be considerable. This uncertainty was estimated by computing the standard deviation of the relative error between the PGV measured and the approximated values when both were available. A final relative error of 56% was obtained in the case where the PGV had to be approximated using the Akkar and Boomer GMPE[81].

**Meteorological data**. Daily precipitations, daily minimum and maximum temperatures were recorded at the SENAMHI meteorological station in Madrigal, 5 km west of the Maca landslide and at the same elevation. Precipitations, at the valley scale, are homogeneous[24] so that meteorological data collected in Madrigal are representative of those at the landslide location. However, although precipitations are known to be similar, no specific studies have compared the temperature between Madrigal and Maca. As the two villages are at the same elevation (3200 m asl), the Madrigal temperature was used as a proxy for the temperature in Maca.

**Effective precipitation**. For this study, the effective precipitation was defined as the daily cumulated precipitation minus evapotranspiration ($ET_0$) that combines two parameters: transpiration and evaporation. The $ET_0$ can be computed in several ways. Due to the limited number of meteorological parameters available, the Hargreaves method[82] was used as it requires only simple meteorological data, such as daily precipitation, daily minimum and maximum temperature[83]. This method has also been demonstrated to be globally valid[49,50]. We used a method described in a crop-evapotranspiration guideline[84], using the temperature and precipitation information measured in Madrigal. $ET_0$ was then multiplied by a factor, Kc, with a value of 0.3 (corresponding to a soil without much vegetation) to obtain the crop evapotranspiration under standard conditions, ETc.

## Data availability
Data are available from the corresponding author upon reasonable request.

## Code availability
Methods are illustrated through text and figure captions and codes are available from the corresponding author upon request.

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

## Acknowledgements

This experiment was funded by an SEG Geophysicists Without Borders grant, a grant from ESA through the Alcantara project "Monitoring and Detection of Landslides from optical Images time-Series" (ESA 15/P26), a CNES/TOSCA grant, and a CNRS/INSU grant. This work was supported by a grant from Labex OSUG@2020 (Investissements d'avenir ANR10 LABEX56). The principal investigator, Noélie Bontemps, is funded by a CNES doctoral fellowship. The authors are grateful for support from the INGEMMET, J. C. Villegas from the Instituto Geofisico del Peru (IGP), the International Plate Boundary Observatory Chile (IPOC, www.ipoc-network.org), Laboratoire International Associé "Montessus de Ballore" (www.lia-mb.net), Central Andean Tectonic Observatory Geodetic Array (CAnTO, http://www.tectonics.caltech.edu/resources/continuous_gps.html), and thank the Centro Sismologico Nacional (CSN, www.csn.uchile.cl) for making the raw GPS data available. Jorge Jara thanks the European Research Council (ERC) for funding under the European Union's Horizon 2020 research and innovation program (grant agreement 758210, project Geo4D). Additional financial support came from the ANR GEO3iLLAB.

## Author contributions

P.L., E.L., E.T. and N.B. installed the seismic and GPS station on the Maca landslide and performed the field experiments. E.T. collected the data every three month in the field. N. B. processed the seismic data and J.J. and N.B. processed the GPS data. N.B. prepared the Figures. P.L., N.B., and E.L. interpreted the data and wrote the paper. P.L. and E.L. designed and coordinated the study.

## Competing interests

The authors declare no competing interests.
