## [Peer Review File · Nature Communications]

Reviewers' comments:

Reviewer #1 (Remarks to the Author):

The authors address an important question of landslide dynamics: the interplay between earthquake shaking and precipitation. Comparing landslide deformation of the Maca landslide in Peru with precipitation and earthquake shaking during a period that includes three wet seasons it is shown that deformation is strongest when precipitation and shaking act together destabilizing the the mass. Moreover the authors use a seismometer to monitor the variation of the seismic velocity in the landslide mass and show that periods of deformation are characterized by velocity at least 0.1% below the maximum velocity attained during the best healed situation.

These are remarkable observation suitable for publication in Nature communications.

I would suggest some rewriting of the manuscript as some statements appear redundant to me whereas other arguments are difficult to follow. Focus on the main observations and discuss their implications one by one.

For example it is hard to understand from paragraph starting on line 107 why undrained loading is excluded. It becomes clear in the supplement, though.

On the technical level I think the seismic investigations included in this article state of the art. A question that was not answered is why the stacking was done only over the time after large events within one day (l386). What is the impact of this restriction? Furthermore, what should be direct waves for single station records? Isn't it rather the autocorrelation of the noise source that should be excluded by leaving out the -0.4 to 0.4 lag time window (l391)? On line 392 you refer to Love- instead of lamb waves, correct? The reference to [25] is useful, that is a nice paper, but not as reference for the stretching technique as on line 388. Also the reference to [52] on line 386 seems misplaced. Ref [52] in methods is the same as [27]

Some minor remarks:

line 140: I think the statement that the "Our measurements show that the combined effect of earthquakes and precipitations on the landslide kinematics is due to damage to surface soil that facilitates the infiltration of water into the ground." is likely but not supported by the data. This appears speculative.

Figure 2: It would be helpful to indicate the times when $dv/v < 1.2\%$ for example with a shading in the background.

Can you tell whether there a difference between healing and draining of the landslide?

What type of bonds are you talking about on lines 175 and 190? Could you specify this?

l375: delete "It was therefore estimated that they could be used to apply the ambient noise interferometry technique⁵⁰."

Why were no autocorrelations (ZZ, EE, NN) used?

Please add units to figure S4 b and c.

It would be really nice to have figures like S1 also for wet and dry seasons in 2017 and 2018.

How representative is figure S4 ($M > 4$ $d < 400$ km) for the present argumentation based on events with $M < 4$ and $d < 50$ km ?

Supplement, Geophysical investigation: "below a few meters" there is the "fully saturated lacustrine layer". What is the impact of precipitation below these few meters, if it is fully saturated. Are the velocity measurements then only affected by these shallow layer.

If Fig. S9e is supposed to indicate that liquefaction by undrained loading is not the mechanism causing displacement of the landslide, then deformation rather than precipitation should be indicated in S9e.

What is the origin of the velocity/density structure used in S9b,c,d

Velocity decrease in S8c does not seem to be 14% below 10m.

Reviewer #2 (Remarks to the Author):

Review of the manuscript "Rain and small earthquakes maintain a slow-moving landslide 1 in a persistent critical state" by Bontemps et al., submitted to Nature Communications

The manuscript investigates a potential combined effect (trigger) of seasonal rainfall and local earthquakes $M > 5$ on a relatively slow-moving (I suggest to check the classification of Hungr et al., 2014...landslides with tens of cm/year cannot be considered slow moving) landslide located in Peru (Maca landslide). The intention of the authors is to focus on a single case study and provide quantitative evidence proving the hypothesis presented qualitatively in previous works, where an increase of landslide activity was observed directly after (or in some cases also several days, months or years after the seismic event). The motivation and the topic are very interesting, the idea to describe the physical processes of a joint earthquake-precipitation effect on a landslide is surely of high scientific interest. However, the paper soon fails in providing robust results that would support such a theory. In some cases, the manuscript is quite confused and lacks of solid background on landslide processes and/or clear explanations on the proposed physical mechanism. Moreover, the

dataset used and the analysis performed are not supporting, in my opinion, the interpretation provided by the authors. For this reason, I suggest rejection of the manuscript. Here below I list only the major issues associated to my criticism.

(1) Some sentences are quite misleading and imprecise. Some examples: L19: the reader understands that in general most of the landslides are triggered by earthquakes, which is not true. L22: progressive weakening of the soil...fracturing of the rock mass. You talk about soil slides? Or rock slides? Having such imprecise sentences in the introduction of a manuscript submitted for publication on Nature is rather astonishing.

(2) L24-L26: the concepts expressed here very simplistic and partially incorrect. It is not very clear which kind of sliding model the authors are presenting (soil slide? rock slide?) High precipitations are often the cause of water table increase and pore pressure changes at depth. The decrease in shear resistance (due to increase in pore pressure) may cause displacements along the sliding surface. Dynamic loading due to earthquake can enhance the effects of this process (undrained loading).

(3) L32-33: In this sentence you extrapolate from a single case (self-referenced, i.e. one of your papers discussing the same landslide you will present later) as a general behavior for landslide. This looks fishy.

(4) In figure 1 is clear that you have information about landslide displacement since 2003 at least. The paper referenced indeed presents long time series of the landslide. Why showing only the last period? I understand that you have the geophysical data starting from 2015, however the relationship between rainfall and displacement for the precedent period could be shown here. Another "peculiarity" visible in Figure 1 is that the landslide is very large (considering the scarp mapped) but the area that is mostly displacing is a smaller portion. So, the depth you present (~40m) is associated to the all slide or just to this faster area?

(5) Your motivation is to provide quantitative proofs of the physical process caused by the combination of earthquake and rainfall, as well as the self-healing. However, the measurements you have about subsurface are only indirect (only geophysical measurements). Without subsurface measurements I think that your hypothesis cannot be ultimately supported for your single case study, nor exported as general behavior for landslides (soil slides? Rock slide? What slides?)

(6) Figure 2. I am not an expert of seismic analysis, however, the dV/V pattern after the M5.5 event in 2016 reminds me the Omori's decay. Are you excluding from your analysis only the main shocks or also the records associated to aftershocks?

(7) Figure 3. This graph is also quite confusing. The peak of the velocity for the year 2016 occurred well before the occurrence of the seismic sequence. But in the way you present the results the intention is to proof that the large displacements are associated to grater seismicity (L92-L95). This is not correct in my opinion

Reviewer #3 (Remarks to the Author):

This is a potentially very interesting article for the Earth science community.

In literature, a precise contribution of rainfall and earthquake component in triggering landslides have never been quantified due to a lack of in-situ data for active landslides. Indeed, current hypotheses as to the physical processes behind how this earthquake-precipitation combination triggers landslides are based only on qualitative observations. Therefore, the treated topic it is still an open challenge for several Earth scientists. Providing clear research on the combined role of rainfall and earthquake forcing in the activation of landslides, describing in details the different phases of the process, it is not an easy job. Until now I never see a clear article showing this. This work by Noélie Bontemps et al. is more than welcome and looking at the results I wish to congratulate with the authors for their effort.

Having said that, unfortunately, there are few critical issues that could mine the entire work at its basis. Here I highlight such points:

- (1) Study area: Maca landslide in Perú. The main critical point here is the fact that only one landslide and one specific location have been considered for the analysis. The authors conclude with these findings: "our results show that the combination of earthquakes and rainfall affects the rigidity of the Maca landslide", "The findings presented here support the idea that medium-intensity earthquakes can accelerate landslides, but that their impact will be greater when precipitations are recorded before or just after the event". Reasonable questions should be: is this a site-specific research? can we generalize such final findings to other study areas? This is the most critical point of the entire work and also one of the major risk treating such a challenging and complex topic. The general impression is that the presented research could be affected by some bias given by the selection of only one landslide. Therefore Nature Communications is not the proper journal, while a more specific natural hazard or geologic engineering journal could be more appropriate.
- (2) One of the results is the fact that measurements show that the combined effect of earthquakes and precipitations on the landslide kinematics is due to damage to surface soil that facilitates the infiltration of water into the ground. Well, this is quite obvious at my eyes. I don't see a great novel finding here.
- (3) I'm not able to find in the text any figure of taken in the field about the described landslide. I strongly recommend adding 2 or 3 when presenting the study area, one at least on GPS locations.

Dear Referees,

Please find the revision of the manuscript NCOMMS-19-18425-T entitled “*Rain and small earthquakes can maintain a slow-moving landslide in a persistent critical state.*” We thank you for your time spent reviewing the manuscript. Your remarks and questions pushed us to clarify the text, to detail some sections as well as to improve some figures. The main changes brought to the main text are: The inclusions of arguments for the non-site specificity of our results in the main text, the clarification of the content when needed, the addition of a paragraph on the model used to estimate the velocity of seismic waves as a function of the saturation. We also changed the limitation of selected earthquakes from a “distance limitation” to a “minimum PGV threshold”, and did some modifications on figures when requested and added several figures in the supplementary information when we thought they could help the global comprehension of the article. Overall, these changes are of a somewhat minor nature. Nevertheless, we feel they certainly improve the paper.

We hope those clarifications help to highlight the main findings of this study, that remained unchanged, that are : (1) we evidence the damage effect of an earthquake on a slow-moving landslide, as well as a recovery processes of this last one, (2) we quantify these two processes thanks to ambient noise monitoring technique, (3) we evidence the combined effect of precipitations and earthquakes on those two processes (damage and healing), (4) we show the importance of small earthquakes on the recovery process, (5) and show the importance of the timing between earthquakes and precipitation on the landslide kinematics and the recovery processes.

Our response letter is organized as follows. First, we summarize the main modifications that have been performed, before providing a detailed point-by-point response to all referees’ comments, in black being their comments and in blue ours.

We hope that all these improvements led to a version of our paper that is now suitable for publication in Nature Communications. We look forward to hearing from you soon.

Best regards,

The authors

Main Modifications

- (1) We answered the question of the site specificity raised by the second and third reviewers, as well as by the editor. The mechanisms showcased here

are neither site specific nor landslide-type specific (rock/soil slide). We base this affirmation on several arguments :

- The characteristics (geology, velocity, mechanism, etc.) of the Maca landslide is similar to a large number of landslides developing in lacustrine deposits (Evans, 1982; Jackson, 2002; Jongmans et al., 2009; Kohv et al., 2010; Roering et al., 2015).
- The landslide kinematics behaves in a similar manner to other soil and rock slides during an earthquake. Only the amplitude of the movement is different from one landslide to the other. (Lacroix et al., 2015)
- The Maca landslide is also similar to a large number of landslides in terms of mechanics: its kinematic is mainly controlled by seasonal precipitations (Coe et al., 2003; Handwerger et al., 2013; Iverson and Major, 1987; Zerathe et al., 2016), with a rainfall threshold required to initiate the motion (Crozier, 1999; Gabet et al., 2004; Iverson and Major, 1987; Van Asch et al., 1999).
- The results on the earthquake/rainfall combination are coherent with previous results on other types of landslides throughout the world (Wistuba et al., 2018; Marc et al., 2015; Durand et al., 2018)
- The mechanism of damage affects all types of earth materials (TenCate, 2011; TenCate et al., 2000). We evidence qualitative effects of damage on a landslide and we quantify it for the Maca landslide. Of course the quantification can be different from one site to another depending on the geology.

We added a paragraph developing these points in the manuscript.

- (2) The comments from reviewer 1 led us to reconsider the limit of 50 km chosen to select the earthquakes having a possible impact on the landslide. The minimum PGV (Peak Ground Velocity) measured among earthquakes within the 50 km limit was 0.01 cm.s^{-1} . A lot of earthquakes with distance larger than 50 km had generated similar or greater PGV at the landslide's location. Hence, all earthquakes with a PGV measured or estimated above 0.01 cm.s^{-1} at the landslide location were taken into account. The minimum PGV taken into account didn't change but we are taking into account a higher number of earthquakes. Even though the number of earthquakes taken into account increased, the results and conclusions on the earthquake/rainfall combination do not change.
- (3) As asked by reviewers 1 and 2, we developed the mechanical aspect in the introduction and clarified our argumentation on the different mechanisms generating the drops of surface wave velocities.
- (4) We slightly changed in a more realistic way the parameters of the model used to simulate the effect of undrained loading on the seismic velocity

variation. We now model the undrained loading as a diminution of 45% of the S-wave velocity over a 2-meter thick layer (based on the paper of Mainsant et al., 2015) at 40 meters depth (instead of an 85% diminution over 50 cm in the previous version). Forty meters depth corresponds to the minimum depth of the sliding surface in the rapid part of the landslide (Zerathe et al., 2016). We modeled a water table of several meters above the sliding surface since one of the requirements for undrained loading is a sliding surface located in a complete saturated soil. We also added a part in the method section about the model that we used, among others, for estimating the velocity of the soil as a function of the saturation.

This new model is more realistic and allows us to conclude that undrained loading, in the sense of an increase pore pressure following co-seismic displacement, is not the mechanism at the origin of the following displacements. Indeed, the investigation of impacted frequencies allows us to conclude that a decrease of S-waves near the sliding surface is definitely not sufficient to explain by itself the strong drop of dv/v between 3 Hz and 8 Hz after the two large earthquakes of magnitude 5 and 5.5 in 2016. In addition, according to this model, a stronger drop of dv/v should have been observed at lower frequencies, which we don't observe.

- (5) We added several figures to the supplementary materials. Two are a close-up on the dv/v and landslide displacement in 2017 and 2018 separately and another one was added to explain the periods used to calculate the displacement and the number of earthquakes in Fig 3. Finally, as proposed by the reviewer 3, we added pictures of the landslide and of the hut containing the GPS and the seismometer.

Reviewer #1:

The authors address an important question of landslide dynamics: the interplay between earthquake shaking and precipitation. Comparing landslide deformation of the Maca landslide in Peru with precipitation and earthquake shaking during a period that includes three wet seasons it is shown that deformation is strongest when precipitation and shaking act together destabilizing the mass. Moreover the authors use a seismometer to monitor the variation of the seismic velocity in the landslide mass and show that periods of deformation are characterized by velocity at least 0.1% below the maximum velocity attained during the best healed situation.

These are remarkable observation suitable for publication in Nature communications.

I would suggest some rewriting of the manuscript as some statements appear redundant to me whereas other arguments are difficult to follow. Focus on the main observations and discuss their implications one by one.

(1) For example it is hard to understand from paragraph starting on line 107 why undrained loading is excluded. It becomes clear in the supplement, though.

We took this comment into account and developed more this section.

On the technical level I think the seismic investigations included in this article state of the art.

(2) A question that was not answered is why the stacking was done only over the time after large events within one day (1386). What is the impact of this restriction?

Thank you for this question that raised a mistake from our part.

Usually, stacks of correlations are made in order to decrease the signal-to-noise ratio. Nevertheless, the counter effect is that the longer the stack over time, the higher the smoothing of the signal, which can mask the abrupt variation of Vs due to an earthquake. For monitoring purposes, stacks are usually made over 24 hours in order to obtain the state of the soil every day (Larose et al., 2015; Mainsant et al., 2012; Voisin et al., 2016). However, being mainly interested in the impacts of earthquakes to the soil rigidity, it was important for us, on the days of strong earthquakes ($PGV > 0.4 \text{ cm.s}^{-1}$), to take into account the state of the soil solely after the earthquake and not an average state of the soil during the whole days.

Therefore, we did try what was exposed at first in the manuscript, i.e., to stack only the hours after the largest event of each day having seismic events with $PGV > 0.4 \text{ cm.s}^{-1}$.

However, this technique was not fully adapted to our problem. Indeed, the considered earthquakes (generated $PGV > 0.4 \text{ cm.s}^{-1}$) sometimes occurred at the end of the day. Hence, only one or two hours were “stacked” and the signal-to-noise ratio dropped drastically. The direct consequence was the appearance of outliers the days of large seismic events due to a complete decorrelation of the green function with regard to the reference, i.e., the average of all the green functions over the whole period of study.

This technique, hence, lost its interest to our eyes and we had chosen to always take into account the 24 hours of each day, even for days with large earthquakes. Nevertheless we forgot to change the text. The consequences of taking 24 hours the days of large seismic events are therefore a possible delay of the dv/v drop for these specific days and a diminution of the measured dv/v drop the exact day of the event due to a smoothing effect.

It is important to say that due to the clipping step in our signal processing (see method) the effect sporadic and of large amplitude sources such as earthquakes is diminished. The sources of the ambient noise are, however, slightly different and thus the green function might be slightly different. These changes are taken into consideration in the uncertainty estimation that is based on the coefficient of correlation in the stretching step. We calculated that the uncertainties are on average

30% higher for the days with large seismic events ($PGV > 0.4 \text{ cm} \cdot \text{s}^{-1}$) compared to other days.

We modified the corresponding section to rectify our mistake.

(3) Furthermore, what should be direct waves for single station records? Isn't it rather the autocorrelation of the noise source that should be excluded by leaving out the -0.4 to 0.4 lag time window (I391)?

Yes, you are right, it was a mistake from our part. We replaced "direct waves" by "autocorrelation of the noise sources." This last one has a duration of $\Delta t = 1/\Delta f$. We decided to remove $2 \times \Delta t$ of the signal (i.e., 0.4 seconds) to be sure that the noise source is not used in the stretching method. We modified the method section.

(4) On line 392 you refer to Love- instead of lamb waves, correct?

Yes, you are right. We rectified the text.

(5) The reference to [25] is useful, that is a nice paper, but not as reference for the stretching technique as on line 388. Also the reference to [52] on line 386 seems misplaced. Ref [52] in methods is the same as [27]

We modified the reference to Mainsant et al. (2015) [25] by a reference to Lobkis and Weaver (2003) and Sens-Schönfelder and Wegler (2006) that are indeed better references for the stretching method. The reference [52] line 386 was indeed misplaced and is, therefore, no more existing in the references of the method section. Thank you for noticing these mistakes.

Some minor remarks:

(6) line 140: I think the statement that the "Our measurements show that the combined effect of earthquakes and precipitations on the landslide kinematics is due to damage to surface soil that facilitates the infiltration of water into the ground." is likely but not supported by the data. This appears speculative.

The damage of the soil suggests an increase in the soil porosity and hence a facilitation of water infiltration. However, our measurements, indeed, don't allow us to do to such a statement. We removed the sentence.

(7) Figure 2: It would be helpful to indicate the times when $dv/v < 1.2\%$ for example with a shading in the background.

Good idea, we did it.

(8) Can you tell whether there is a difference between healing and draining of the landslide?

The healing phase is well documented in laboratory observations (TenCate, 2011) as well as in field observations (Brenquier et al., 2008; Gassenmeier et al., 2016; Hobiger et al., 2012; Richter et al., 2014; Wang et al., 2007; Wegler et al., 2009).

Observations and models (empirical (Gassenmeier et al., 2016; Richter et al., 2014) or based on mechanical and chemical processes at the granular scale (Snieder et al., 2016)) find that the healing follows a logarithmic trend with time, with the specificity that it should tend toward 0 for an infinite time to be realistic (and not toward infinity as a normal logarithmic curve). Hence, we can confirm that it is well a healing process that we observe in August 2016, due to the shape of the dv/v after the earthquake. However, during wet seasons we can't dissociate the two processes as we don't observe the logarithmic recovery. Moreover, the presence of water possibly impact the recovering process, hence making the separation of healing and drainage even more complicate.

We mentioned that we are not able to distinguish between drainage and healing processes during wet seasons in the manuscript (l 153).

(9) What type of bonds are you talking about on lines 175 and 190? Could you specify this?

We are talking about bonds between the elements of the highly unconsolidated granular material (clay, shale and siltstone (Zerathe et al., 2016)), which include weak chemical bonds and capillary bonds (Chang and Woods, 1992; Snieder et al., 2016).

We clarified it in the text.

(10) l375: delete "It was therefore estimated that they could be used to apply the ambient noise interferometry technique50."

We deleted it as suggested.

(11) Why were no autocorrelations (ZZ, EE, NN) used?

We didn't use the autocorrelation for two reasons. First of all, when doing autocorrelation, we also auto-correlate the electronic noise from the seismometer. This effect is smaller when correlating two different channels but in general strong enough to affect the autocorrelations. This will depend on the digitizer performance.

The second reason is that the whitening step has to be removed from the processing when correlating the same channels (De Plaen et al., 2019), which becomes a problem when the frequency content of the noise is not completely stable with time (Mikesell et al., 2015; Zhan et al., 2013). Indeed, it has been shown that when using the stretching method, frequency variations of the ambient noise could generate spurious variations of the dv/v (Zhan et al., 2013). The whitening of the signal allows to diminish the influence of frequency variations by equalizing the frequency content of the ambient noise.

We know from a power spectral density analysis on our data that the 4th of June 2016, a new source of noise started to illuminate with a higher amplitude the seismic station around 6 Hz to 7 Hz. We took care to verify when doing single station cross-correlation that this variation in the ambient noise frequency content did not influence the dv/v when filtering between 3 Hz and 8 Hz and when using the whitening step.

However, when doing single station autocorrelation (ZZ, NN, EE), the whitening step is not possible anymore and we do observe a dv/v drop starting on the 4th of June 2016 (see figure below that shows actual artifacts).

Therefore, single station autocorrelation is not reasonable in our case, at least in the 3 to 8 Hz frequency band.

Figure : Relative seismic changes to the velocity of the material determined by comparing daily seismic noise correlograms in the 3–8 Hz frequency range. The same steps as in the paper were performed except that we used single station autocorrelation (ZZ, EE, NN) instead of single station cross-correlation (ZE, ZN, NE) and that we didn't whiten the signal before correlation.

(12) Please add units to figure S4 b and c.

We added them.

(13) It would be really nice to have figures like S1 also for wet and dry seasons in 2017 and 2018.

We added them to the supplementary material (Fig S4 and S5).

(14) How representative is figure S4 ($M > 4$ $d < 400$ km) for the present argumentation based on events with $M < 4$ and $d < 50$ km ?

GMPEs were used only in case of saturated signal at the Maca station and hence were mainly not used for $M < 4$ and $d < 50$ km, for which we could measure the real PGV. Nevertheless this question is completely relevant as, indeed, some earthquakes with magnitude lower than 4 did saturate the seismic signal. Our previous uncertainty of estimated PGV was therefore not representative of all earthquakes saturating the signal.

To correct this, we chose to enlarge the initial limits of our comparison by taking into account earthquakes with magnitude > 3.5 instead of > 4 .

We kept the maximal distance at 400 km, as previously, for the reason that only one earthquake above this threshold needed an estimation of the PGV. In addition, this earthquake's location was 657 km away from the landslide, i.e., far beyond the limits of validity of the two GMPEs, and did not impact the dv/v .

We want to emphasize here, as mentioned previously, that it appeared to us that the limit of 50 km wasn't the best proxy to select earthquakes that could have possibly impacted the landslide kinematic. Hence we decided to consider all earthquakes that had a PGV (measured or estimated) higher than $0.01 \text{ cm}\cdot\text{s}^{-1}$. This new limit consists in the minimum PGV measured for earthquakes registered in the IGP catalog within the local seismicity (i.e., a distance inferior to 50 km from the landslide). This increases the number of earthquakes taken into account from 178 to 348. The results and main conclusions of the article, however, remained unchanged: the year 2018 had still more earthquakes recorded at the end of the rainy seasons than in 2017, and we still argue that the landslide was kept in a critical state in 2018 by small magnitude earthquakes coupled with a high saturated soil, preventing the landslide from increasing its rigidity.

(15) Supplement, Geophysical investigation: "below a few meters" there is the "fully saturated lacustrine layer". What is the impact of precipitation below these few meters, if it is fully saturated? Are the velocity measurements then only affected by these shallow layer.

The dv/v being sensitive to depths of approximately $1/3$ of the wavelength, the velocity measurements are sensitive to the first 40 meters in the 3-8 Hz frequency range. As long as the landslide is in motion or that the soil is damaged by earthquake shaking, the dv/v might be impacted in saturated soil. Moreover, the assumption of the fully lacustrine deposits, here, is only valid because the geophysical investigations were made at the very end of the rainy season. Hence we assume that the lacustrine layer, normally not very permeable, had time to be filled via a network of fractures and fissures and thanks especially to the shallower water table. However, this geophysical investigation is a snapshot of the velocity in the soil at a specific moment. Drainage and fluid circulation will still occur and hence the saturation of the lacustrine layer will change. In addition, shearing of the sliding surface at depth will also influence the dv/v . In conclusion, the dv/v , in the 3-8 Hz frequency range, will also be sensitive to variations in rigidity and density below the shallow water table.

We specified that the assumption of a fully saturated layer was only valid at a specific time and used to calibrate the Biot-Gassmann's equations (see the answer to the question 16). To this purpose we added the following paragraph in the geophysical investigation in the SI:

"Active seismic acquisitions only correspond to snapshots of the velocity of P and S waves at a given time. Given that these acquisitions were realized at the very end of the rainy season, we assume, for simplification, that P and S wave velocities of the

second layer correspond to a completely saturated lacustrine deposit layer. This assumption, then allows us to calibrate the parameters in Biot-Gassmann's equations (see method section) to retrieve the velocity of P and S waves as a function of the saturation.”

(16) If Fig. S9e is supposed to indicate that liquefaction by undrained loading is not the mechanism causing displacement of the landslide, then deformation rather than precipitation should be indicated in S9e.

We are sorry, we are not sure to understand this comment. We assume your comment certainly comes from the fact that most (if not all) of the studies focusing on the undrained loading on landslides are mentioning triggering of rapid landslides. Co-seismic displacements of the landslide generated by earthquakes accelerations lead to grain crushing at the basal interface, hence creating pore pressure build-up, which, if not counterbalanced, might lead to rapid flows (Wang et al., 2007). Models and observations have shown that depending on soil properties such as grain size, permeability and hydraulic diffusivity, slow moving landslide motion can be stabilized or diminished by mechanisms such as pore pressure feedback, dilation and the development of deformation cracks (Iverson, 2005; Moore and Iverson, 2002; Schulz et al., 2009). Rapid and catastrophic triggering of landslides is thus prevented. These mechanisms are the ones mentioned in order to explain why slow-moving landslide don't transform into rapid landslides when influenced by pore pressure variation induced by precipitations (Handwerker et al., 2013; Schulz et al., 2009).

The main point of this figure was to show either that undrained loading didn't occur at all or that undrained loading on its own couldn't have generated the dv/v we measured. Then damage of the soil would be mostly at the origin of the variations that we observe. The displacement on its own cannot say what mechanism is at its origin. However, we changed the figure and displayed both the dv/v at a frequency range of 0.2 to 2 Hz and the displacement. In addition, we zoomed in on the two earthquakes as for the figure 1 in the SI. Finally, we added to the figure S14 (old S9) our measurements of dv/v at different frequencies, showcasing the difference between the undrained loading prediction and the real observations.

(17) What is the origin of the velocity/density structure used in S9b,c,d

In August, when the second large earthquake occurred, the rainy season was over since a couple of months already. Hence the velocity model with a completely saturated soil is not valid anymore. We calculated the velocity of P and S waves in the soil as a function of saturation from the Biot-Gassmann equations (see the added section in the methodology). Hence, the velocity model just before the August earthquake is based on the velocity of lacustrine deposits saturated at 25%, representing a non-saturated soil. Thereafter, we modeled a saturated layer of 5

meters above the sliding surface that is located around 40 meters in order for the landslide to be in conditions where undrained loading could actually occur.

Regarding the model in case of undrained loading, we chose to slightly modify it compared to what we have done in the previous version of the manuscript. We now model the liquefaction of a layer of 2 meters instead of only 50 cm to be in accordance with other studies where the thickness of the shearing zone is between 1 and 2 meters (Jongmans et al., 2009; Kane et al., 2001; Mainsant et al., 2012; Simeoni and Mongiòvi, 2007). Then we decided to decrease the velocity of only 45 % instead of 85% based on the study by Mainsant et al. (2012), which is the only study quantifying the variation of Vs due to a landslide fluidization. In this last study, 45% of decrease of S waves were sufficient for the landslide to fluidize and to be triggered rapidly.

The density and Vp shouldn't change that much considering that the sliding zone is completely saturated. We therefore chose to let them stable.

We described a bit more the process we used to estimate the velocity of P and S waves as a function of the saturation in the method section. In addition, we developed more the undrained loading model in the SI.

(18) Velocity decrease in S8c does not seem to be 14% below 10m.

The figure 8c (current figure S13) seems correct to us. Indeed, 14% of 350 m/s is equal to 49 m/s. Hence a decrease of 14% from 350 m/s leads to a velocity of ≈ 300 m/s.

Reviewer #2:

Review of the manuscript "Rain and small earthquakes maintain a slow-moving landslide 1 in a persistent critical state" by Bontemps et al., submitted to Nature Communications

The manuscript investigates a potential combined effect (trigger) of seasonal rainfall and local earthquakes $M > 5$ on a relatively slow-moving (I suggest to check the classification of Hungr et al., 2014...landslides with tens of cm/year cannot be considered slow moving) landslide located in Perú (Maca landslide).

The reviewer is right, the landslide's velocity being below 1.6 m/yr during the studied period, it can be classified in the very slow-moving landslide category (Cruden and Varnes, 1996). However, over the 28 years period (1986-2013), the landslide moved by 50 m in its fastest zone (Bontemps et al., 2018), and at some period of his life the landslide motion was much higher than 1.6 m.yr^{-1} (as in 2011-2012 where the displacement was 7 m over a year (Zerathe et al., 2016)). Therefore, the landslide could be considered as sometimes slow and sometimes very slow. We kept the term of slow-moving landslide throughout the manuscript for simplicity, but in the context and experimental setup we characterized it as "slow- to very slow-moving landslide".

The intention of the authors is to focus on a single case study and provide quantitative evidence proofing the hypothesis presented qualitatively in previous works, where an increase of landslide activity was observed directly after (or in some cases also several days, months or years after the seismic event). The motivation and the topic are very interesting, the idea to describe the physical processes of a joint earthquake-precipitation effect on a landslide is surely of high scientific interest. However, the paper soon fails in providing robust results that would support such a theory. In some cases, the manuscript is quite confused and lacks of solid background on landslide processes and/or clear explanations on the proposed physical mechanism. Moreover, the dataset used and the analysis performed are not supporting, in my opinion, the interpretation provided by the authors. For this reason, I suggest rejection of the manuscript. Here below I list only the major issues associated to my criticism.

(1) Some sentences are quite misleading and imprecise. Some examples: L19: the reader understands that in general most of the landslides are triggered by earthquakes, which is not true.

We agree that precipitation is also an important factor triggering rapid landslides or impacting the kinematics of slow-moving ones. We slightly modified the introduction (line 31 to line 33, in the document with modifications) in order to make it clearer that we focus only on earthquake triggering mechanisms, while considering also precipitations as a mechanism impacting landslide kinematics.

(1 bis) L22: progressive weakening of the soil...fracturing of the rock mass. You talk about soil slides? Or rock slides? Having such imprecise sentences in the introduction of a manuscript submitted for publication on Nature is rather astonishing.

The landslide of Maca has been categorized by Zerathe et al. (2016) as a clay/silt compound slide with a rupture surface of uneven curvature. This would make it a soil slide. The category of the landslide wasn't mentioned in the presentation section of the landslide. This was an oversight from our part and therefore mentioned it to the text.

(2) L24-L26: the concepts expressed here very simplistic and partially incorrect. It is not very clear which kind of sliding model the authors are presenting (soil slide? rock slide?) High precipitations are often the cause of water table increase and pore pressure changes at depth. The decrease in shear resistance (due to increase in pore pressure) may cause displacements along the sliding surface. Dynamic loading due to earthquake can enhance the effects of this process (undrained loading).

We now precise the type of slide (see our previous answer). The processes studied here are, however, not specific to one type of landslide. Indeed undrained loading affects both rockslide (King et al., 1989; Sassa et al., 2007; Xu et al., 2012) and soil

slides (Gratchev et al., 2006; Sassa et al., 2004). We added a sentence on this in the text. Perhaps some explanations were missing on the fact that undrained loading shouldn't lead every time to rapidly triggered landslides. Models and observations have shown that depending on soil properties such as grain size, permeability and hydraulic diffusivity, slow moving landslide motion can be stabilized or diminished by mechanisms such as pore pressure feedback, dilation and the development of deformation cracks (Iverson, 2005; Moore and Iverson, 2002; Schulz et al., 2009). Rapid and catastrophic triggering of landslides is thus prevented.

Undrained loading was studied as a possible mechanism reactivating the landslide in this article. However, we show that it does not explain the observations well.

We do agree that high precipitation is one of the main forcing of landslide kinematics/ triggering. As mentioned previously, we added precipitations as one of the main forcing in the introduction, even though the main goal of the article is to focus on earthquake triggering mechanisms.

(3) L32-33: In this sentence you extrapolate from a single case (self-referenced, i.e., one of your papers discussing the same landslide you will present later) as a general behavior for landslide. This looks fishy.

To our knowledge, this is the only paper showcasing the long-term impact of earthquakes on slow-moving landslides. This is why this is the only paper cited. Now, we also explain why this case-study can be extrapolated to a more general situation (see our main modifications section).

(4) In figure 1 is clear that you have information about landslide displacement since 2003 at least. The paper referenced indeed presents long time series of the landslide. Why showing only the last period?

I understand that you have the geophysical data starting from 2015, however the relationship between rainfall and displacement for the precedent period could be shown here.

We indeed have GPS information since 2001 (Zerathe et al., 2016) and satellite measurements since 1986 (with much larger uncertainties). However the measurements were really sparse between 2001 and 2011 (only 3 GPS campaigns and 6 satellite acquisition), and the satellite acquisitions are also sparse over the whole period 1986-2014 (16 satellite acquisition, Bontemps et al., 2018). This precludes their use for yearly displacement estimations. However, we are doing GPS campaigns every 3 months or so since November 2011 (Zerathe et al., 2016). We thought, at first, about also representing the years 2012 and 2013 in the figure 3. However, the year 2012 was an exceptional year in terms of precipitation where the landslide underwent 7 meters of displacement, i.e., almost seven times its average yearly displacement. Plotting this exceptional year on the figure would have hidden combined effect of earthquakes and precipitation on more normal rainy seasons.

Secondly, following the intense activity of the Sabancaya Volcano in 2013, the IGP densified his seismometer network in the Colca Valley. The number of earthquakes detected by the IGP, hence, increased after 2013 and the magnitude of the smallest detected events decreased from $\approx Ml = 4$ to $\approx Ml = 3$. This difference complicates the yearly comparison of the number of earthquakes possibly impacting the landslide before/after 2013.

In conclusion, we decided to only take into account data that we could compare (from 2014 to 2018) and not to add more information on this figure. We added the two reasons explained here in the SI, on a part specifically written to explain the creation of the figure 3 and the choice of the years displayed.

(4 bis) Another “peculiarity” visible in Figure 1 is that the landslide is very large (considering the scarp mapped) but the area that is mostly displacing is a smaller portion. So, the depth you present (~40 m) is associated to the all slide or just to this faster area?

Yes, the ~40 meters mentioned in the main text is for the area that we are studying, i.e., the faster area, also corresponding to the largest active block of the landslide. We modified the text for it to be clearer: “. In the fastest part of the landslide, the sliding surface is located at least 40 m down from the surface, within the lacustrine deposits.”

(5) Your motivation is to provide quantitative proofs of the physical process caused by the combination of earthquakes and rainfall, as well as the self-healing. However, the measurements you have about subsurface are only indirect (only geophysical measurements). Without subsurface measurements I think that your hypothesis cannot be ultimately supported for your single case study, nor exported as general behavior for landslides (soil slides? Rock slide? What slides?)

Subsurface measurements could indeed give interesting and important information at depth. However, they correspond only to very local information. On the contrary, geophysical measurements scan a higher volume and can be easily displaced in order to see another part of the landslide.

In addition, geophysical measurements such as ambient noise interferometry used here, have the non-negligible advantage to give consistent information of the subsurface with time. This would not be feasible with subsurface measurements in such a landslide: a borehole would be destructed in a couple of months due to the velocity and shearing of the landslide.

Here we present unique data of a very slow-moving landslide under earthquake and precipitations forcings, allowing to evidence damage, healing and soil recovery processes. It is true that the quantification of such processes might be different depending on the geology of the landslide. However, the processes are valid for all types of landslides, and the main point of this paper is to highlight these processes.

(6) Figure 2. I am not an expert of seismic analysis, however, the dv/v pattern after the M5.5 event in 2016 reminds me the Omori's decay. Are you excluding from your analysis only the main shocks or also the records associated to aftershocks?

We do remove the hours before large earthquakes and clipped the signal to remove the large amplitude of earthquakes signals. However, these steps don't remove the impact of the seismic shaking to the soil in itself, which is what we measure via the variation of the velocity of surface waves in the soil. Hence, the dv/v calculated still represent the state of the soil after each seismic event, main shock or aftershock similarly.

In August 2016, the main shock strongly damaged the soil. This is confirmed by the strong decrease of the dv/v . As shown by many authors on stable soils (Gassenmeier et al., 2016; Richter et al., 2014; Snieder et al., 2016; TenCate, 2011), the healing of the soil after a large shaking event, also called slow dynamic, is well represented by a logarithmic function with time. If, after a main shock, aftershocks continue to damage the soil, other drops of dv/v (i.e., negative variations) will be measured, as observed and modeled by Gassenmeier et al. (2016). However, if the aftershocks don't have any effects on the soil rigidity, the healing of the soil will continue normally and the dv/v variation observed should be positive.

(7) Figure 3. This graph is also quite confusing. The peak of the velocity for the year 2016 occurred well before the occurrence of the seismic sequence. But in the way you present the results the intention is to proof that the large displacements are associated to grater seismicity (L92-L95). This is not correct in my opinion

If we understand well your concern, the misunderstanding comes from the fact that, in 2016, two different seismic sequences occurred, one in February and one in August. The August sequence is the largest one, but the landslide velocity was higher in February-March due to the combination with the rainy season.

The goal of the Figure 3 is to showcase the combined effect of the earthquake and precipitation on the landslide kinematics at a yearly time scale. We therefore agree that the August crisis during the dry season could indeed be confusing.

In this figure, we decided to take only displacements that started during the wet season up to the first steady state of the landslide. This removes the displacement following the August 2016 earthquake. Regarding the number of earthquakes, all events with a magnitude above $0.01 \text{ cm}\cdot\text{s}^{-1}$ were counted within the periods of interest. We added a figure in the SI to showcase the selected periods.

Overall, the main change in the figure 3 is the number of earthquakes that is taken into account. More events were selected as the threshold for counting earthquakes passed from a limit distance (50 km from the landslide) to a threshold on the minimum PGV taken into account (0.01 cm/s). The results of this figure remain

the same: we observe a clear combination between earthquake activity and precipitation impacting the kinematics of the landslide.

Reviewer #3:

This is a potentially very interesting article for the Earth science community. In literature, a precise contribution of rainfall and earthquake component in triggering landslides have never been quantified due to a lack of in-situ data for active landslides. Indeed, current hypotheses as to the physical processes behind how this earthquake-precipitation combination triggers landslides are based only on qualitative observations. Therefore, the treated topic it is still an open challenge for several Earth scientists. Providing clear research on the combined role of rainfall and earthquake forcing in the activation of landslides, describing in details the different phases of the process, it is not an easy job. Until now I never see a clear article showing this. This work by Noélie Bontemps et al. is more than welcome and looking at the results I wish to congratulate with the authors for their effort.

Having said that, unfortunately, there are few critical issues that could mine the entire work at its basis. Here I highlight such points:

(1) Study area: Maca landslide in Peru. The main critical point here is the fact that only one landslide and one specific location have been considered for the analysis. The authors conclude with these findings: "our results show that the combination of earthquakes and rainfall affects the rigidity of the Maca landslide", "The findings presented here support the idea that medium-intensity earthquakes can accelerate landslides, but that their impact will be greater when precipitations are recorded before or just after the event".

Reasonable questions should be: is this a site-specific research? Can we generalize such final findings to other study areas? This is the most critical point of the entire work and also one of the major risk treating such a challenging and complex topic. The general impression is that the presented research could be affected by some bias given by the selection of only one landslide. Therefore Nature Communications is not the proper journal, while a more specific natural hazard or geologic engineering journal could be more appropriate.

Indeed, these measurements were only made on one landslide due to the large effort required to install and maintain such monitoring over many years on the Altiplano. These data are hence completely new, and of course, raising the question of the site specificity.

First of all, the landslide of Maca behaves in a similar manner to other landslides during an earthquake (both in lacustrine and in rock sediments). Only the amplitude of the kinematic response is different from one landslide to another, due to differences of geology and distance to the source (Lacroix et al., 2015).

The landslides in this valley are also similar in terms of geology, thickness and velocity to other landslides developing in other areas of the world, for example in the glacio-lacustrine clays in the Trièves area, French Alps (Jongmans et al., 2009), in the lacustrine deposits in the California Eel Valley (Roering et al., 2015), in British Columbia, Canada (Evans, 1982; Jackson, 2002) or in the Baltic region (Kohv et al., 2010), among others.

The Maca landslide is also comparable to other landslides in terms of mechanism. Its kinematics is mainly controlled by seasonal rainfall and it requires a certain amount of cumulative precipitation to initiate a movement (Zerathe et al., 2016). This has been observed in many other landslides all over the world, making this process independent of the geology considered (Bogaard and Greco, 2016; Guzzetti et al., 2012; Iverson and Major, 1987; Marc et al., 2018; Scheevel et al., 2017; Terlien, 1998). The value of the threshold itself, however, can be geology dependent.

Here, we evidence some processes of earthquake-precipitation combination on one landslide. Of course, quantitative differences will exist between different sites, depending on the geology, the depth of the landslide, pore saturation, etc. Nevertheless, in a qualitative manner, damage mechanisms aren't site-specific and geology specific when submitted to earthquake shaking (TenCate, 2011; TenCate et al., 2000).

Eventually, our data explains well observations of (1) rainfall threshold variation after earthquakes (Bontemps et al., 2018; Lin et al., 2004; Zhang and Zhang, 2017), (2) increased rates of rapidly triggered landslides for several years after large events (Marc et al., 2015), (3) impact of low magnitude earthquakes to the kinematic of slow-moving landslide when combined with higher precipitation (Wistuba et al., 2018) and (4) the higher rates of rockfalls in a volcanic dome when the seismicity is higher (Durand et al., 2018).

We think that these arguments can help to see that, even though we have data on only one site, these results are non-site specific and are of interest for the whole landslide and geomorphology community. We develop the similitude of the Maca landslide in comparison to other landslides mentioned here in this new version of the manuscript in the context and experimental setup section as well as in the conclusion.

(2) One of the results is the fact that measurements show that the combined effect of earthquakes and precipitations on the landslide kinematics is due to damage to surface soil that facilitates the infiltration of water into the ground. Well, this is quite obvious at my eyes. I don't see a great novel finding here.

The observation of a combined effect of earthquakes and precipitations on landslide triggering/kinematics is not new (Marc et al., 2015; Wistuba et al., 2018). All these studies hypothesized that damage was a potential process to explain these observations. Still, the damage to the soil has never been actually measured. The

main novelties of this study are that (1) we evidence this damage effect as well as a recovery process, (2) we quantify these two processes, (3) we evidence the combined effect of precipitations and earthquakes on those 2 processes, (4) we show the importance of small earthquakes on the recovery process, (5) and show the importance of the timing between earthquakes and precipitation on the landslide kinematics and the recovery processes. The novelty is therefore not only on evidencing the damage process.

All these novelties are already mentioned in the abstract and conclusion, except for the point (5) previously mentioned on the temporality between forcings, which had been omitted in the abstract. Hence we completed the abstract.

(3) I'm not able to find in the text any figure of taken in the field about the described landslide. I strongly recommend adding 2 or 3 when presenting the study area, one at least on GPS locations.

This is a good idea, we added some pictures from the landslide and from the hut with the GPS and the seismometer. We also mentioned the paper of Zerathe et al. (2016) where other pictures of the landslide can be found.

REVIEWERS' COMMENTS:

Reviewer #1 (Remarks to the Author):

The authors revised the manuscript in response to the numerous comments by the reviewers and replied in length.

The key message of the manuscript is clear: the landslide activity is stronger in 2016 when the seismic activity was strong in the rainy season. Later seismic events in the dry season had smaller effects despite stronger shaking. I think this is worth publishing in NatComm. However, it is a case study.

The revision clarified the issues with the previous version. In the reworked section in the supplement about liquefaction at 40m depth, I see some new issues that can probably be fixed easily.

It is not clear how the dv/v values in Fig. S14 top right are measured, what is the averaging time window used to obtain these values. So the figure is not reproducible, which it should be. In the caption Fig S14 the authors talk about a change in the reference required by a sudden frequency change in the ambient noise. This is confusing, was not mentioned in the earlier version (nor is it mentioned in the Method section) and I could not understand how this is handled and what the consequences are. This has to be clarified.

The conclusion drawn from fig S14 that liquefaction cannot account for the observed dv/v strongly depends on the fact that the interface is at 40m depth. I guess this estimate involves some uncertainty especially since the fast movement involves only part of the landslide. What if the sliding interface in this part was at 20m? Can this be excluded from the dynamics of this landslide? Could liquefaction still be excluded with a similar velocity model that has the low velocity layer at 20m depth?

Reviewer #2 (Remarks to the Author):

Dear authors,

thanks for your replies and the revised version of the manuscript. You have done a good work in providing more details for a better evaluation of your hypothesis, analyses, and results.

Despite such improvements, however, I do not change my general impression on this work. The data and analyses presented do not lead to conclusive results that would provide major impact on the actual level of knowledge of the topic. I believe that presenting a single landslide (site specific conditions), only 3-4 years of monitoring observations, and a very little number of earthquake sequences (3, out of which only one seems to show clear evidence) is not enough to provide strong general conclusions, required to deserve publication in a Nature journal, on the processes associated to the (potential) combined effects of rainfall and earthquakes on landslide kinematics. Different would have been to have a longer (10 years or more), homogeneous, time series of both geophysical and geodetic observations for a single landslide AND/OR the same observations and analyses on

multiple landslides located in different geological and tectonic settings. Having such a short observation windows is a major limitation and could lead to important biases in your interpretations.

For this reason, I still suggest rejection of this manuscript for publication in Nature Communications although I encourage authors to submit the manuscript to a more specific journal in the field of engineering geology.

Reviewer #3 (Remarks to the Author):

The authors provided a detailed reply to my comments; I'm now convinced, therefore from my side i suggest to accept the work.

Reviewer #1 (Remarks to the Author):

The authors revised the manuscript in response to the numerous comments by the reviewers and replied in length.

The key message of the manuscript is clear: the landslide activity is stronger in 2016 when the seismic activity was strong in the rainy season. Later seismic events in the dry season had smaller effects despite stronger shaking. I think this is worth publishing in NatComm. However, it is a case study.

1 - The revision clarified the issues with the previous version. In the reworked section in the supplement about liquefaction at 40m depth, I see some new issues that can probably be fixed easily.

It is not clear how the dv/v values in Fig. S14 top right are measured, what is the averaging time window used to obtain these values. So the figure is not reproducible, which it should be.

Indeed some information was missing in order to make this figure reproducible. We used the same parameter as presented in the method section that we slightly modified in order to be understandable for different frequency bands. The averaged time window is now described as a function of Δt , the duration of the autocorrelation of the noise sources in the method section. This duration depends on the studied frequency band: $\Delta t = 1/\Delta f$. Hence, the time window used for the stretching method corresponds to $[14 \Delta t : 2\Delta t]$ $[2\Delta t : 14 \Delta t]$.

Then, instead of looking only at the difference between the day prior and the one following the two large earthquakes, like in the last version of the manuscript, we decided to show here the difference between the three days before and the three days after the two earthquakes. This allows to limit the impact of the aftershocks that will slightly change the noise source structure and hence diminish the quality of the dv/v just after large earthquakes. This way, we are confident that we can compare the drops in the different frequency bands for both $M_I > 5$ earthquakes. We explain this in the legend of the Supplementary figure 14.

2 - In the caption Fig S14 the authors talk about a change in the reference required by a sudden frequency change in the ambient noise. This is confusing, was not mentioned in the earlier version (nor is it mentioned in the Method section) and I could not understand how this is handled and what the consequences are. This has to be clarified.

The mention of this source change was an oversight from our part in the first version of the manuscript that we felt important to correct in the second new version for reproducibility reasons. However, this change of source has no impact on our observations and analysis: indeed this change in the ambient noise seems to be localized to certain high frequencies (around 10 Hz), whereas we limit our analysis to frequencies below 8 Hz. This change in sources is possibly due to electronic noise, either from our instruments or from a new electronic device installed near the seismometer (including most probably radio signals).

The consequences of such frequency change in the noise are, in our case, the generation of noisier dv/v time series. The whitening of the signals in the processing of the signal becomes even more important to counteract such issues. In our case, the whitening of the signals in the frequency band below 8 Hz is sufficient to keep a good signal-to-noise ratio in the dv/v , similar throughout the whole studied frequency bands. Hence, below 8 Hz it is possible to keep the same reference taking into account the three years of acquisition and we could have plotted the drop of dv/v of the February 2016 earthquake in the supplementary figure 14 in our last version of the supplementary material. This part was a mistake from our part.

Indeed, two solutions were possible for us in this case: (1) not to study above 8 Hz or (2) study above 8 Hz but not plotting the dv/v drop for the February 2016 earthquake for frequency bands taking into consideration frequencies above 8 Hz. Indeed, only the period between February and June 2016 is noisier when considering frequencies above 8 Hz.

At first, we chose the second option and then changed our minds as it became too complicated to explain the issue without generating confusion. However, with regards to the last version of the supplementary material, it seems that we got mixed up in our explanation and figures when doing the modification.

Here, we propose to study the dv/v only up to 8 Hz to avoid the influence of new electronic noise starting on the 4th of June 2016. This permits the addition of the dv/v drop of the February 2016 earthquake that was missing in the Supplementary figure 14 and to withdraw information about electronic noise and ambient noise frequency variation in the text. This is only possible due to the fact that it doesn't influence the signal below 8 Hz thanks to the whitening operation. This would strongly help the comprehension of the principal conclusions of the manuscript.

3 - The conclusion drawn from fig S14 that liquefaction cannot account for the observed dv/v strongly depends on the fact that the interface is at 40m depth. I guess this estimate involves some uncertainty especially since the fast movement involves only part of the landslide. What if the sliding interface in this part was at 20m? Can this be excluded from the dynamics of this landslide? Could liquefaction still be excluded with a similar velocity model that has the low velocity layer at 20m depth?

We also wondered what should be the impact on the model if the liquefaction occurred at a shallower layer. In the figure presented below, the model for the liquefaction of a 1-meter layer at 20 meters depth was added to the supplementary figure 14 for comparison. The dv/v drop becomes more important and impacts higher frequencies. This is even more contradictory to our measurements on the landslide, and hence, was not considered as a possibility here. We added the following sentence in the supplementary note 4: "The model tells us that, if undrained loading occurs at lower depth (e.g., 20 meters), the expected dv/v drop together with the impacted frequencies will be higher. This would be even more in contradiction to our measurements and hence, the existence of a layer impacted by undrained loading at lower depth seems not in accordance with our observations."

We didn't add the 20-meter depth model to the supplementary figure 14 to keep it simple.

Figure 1: dv/v expected to be observed by the seismometer as a function of frequency for the liquefaction of a 2 m layer representing the sliding surface at 40 m depth (continuous line) and at 20 m depth (dashed line). Times series of the dv/v were first calculated in the different frequency bands (0.2-2Hz, 2-4 Hz, 4-6 Hz, 6-8 Hz and 3-8 Hz). The time window used for the stretching is the following are the following: $[14 \Delta t : 2\Delta t]$ $[2\Delta t : 14 \Delta t]$, with $\Delta t = 1/\Delta f$. The drop of dv/v for the two large events ($M_L > 5$) is calculated by subtracting the average of the dv/v of the three days following each earthquake to the three days preceding them.

Reviewer #2 (Remarks to the Author):

Dear authors,

thanks for your replies and the revised version of the manuscript. You have done a good work in providing more details for a better evaluation of your hypothesis, analyses, and results.

Despite such improvements, however, I do not change my general impression on this work. The data and analyses presented do not lead to conclusive results that would provide major impact on the actual level of knowledge of the topic. I believe that presenting a single landslide (site-specific conditions), only 3-4 years of monitoring observations, and a very little number of earthquake sequences (3, out of which only one seems to show clear evidence) is not enough to provide strong general conclusions, required to deserve publication in a Nature journal, on the processes associated to the (potential) combined effects of rainfall and earthquakes on landslide kinematics. Different would have been to have a longer (10 years or more), homogeneous, time series of both geophysical and geodetic observations for a single landslide AND/OR the same observations and analyses on multiple landslides located in different geological and tectonic settings. Having such a short observation windows is a major limitation and could lead to important biases in your interpretations.

For this reason, I still suggest rejection of this manuscript for publication in Nature Communications although I encourage authors to submit the manuscript to a more specific journal in the field of engineering geology.

Reviewer #3 (Remarks to the Author):

The authors provided a detailed reply to my comments; I'm now convinced, therefore from my side i suggest to accept the work.